# Evaluation of Quality and Bone Microstructure Alterations in Patients with Type 2 Diabetes: A Narrative Review

**DOI:** 10.3390/jcm11082206

**Published:** 2022-04-14

**Authors:** José Ignacio Martínez-Montoro, Beatriz García-Fontana, Cristina García-Fontana, Manuel Muñoz-Torres

**Affiliations:** 1Department of Endocrinology and Nutrition, Virgen de la Victoria University Hospital, Instituto de Investigación Biomédica de Málaga (IBIMA), Faculty of Medicine, University of Malaga, 29010 Malaga, Spain; joseimartinezmontoro@gmail.com; 2Bone Metabolic Unit, Endocrinology and Nutrition Division, University Hospital Clínico San Cecilio, 18016 Granada, Spain; 3Instituto de Investigación Biosanitaria de Granada (Ibs. GRANADA), 18012 Granada, Spain; 4Centro de Investigación Biomédica en Red Fragilidad y Envejecimiento Saludable (CIBERFES), Instituto de Salud Carlos III, 28029 Madrid, Spain; 5Department of Medicine, University of Granada, 18016 Granada, Spain

**Keywords:** type 2 diabetes mellitus, bone fragility, fracture risk, bone structure, bone quality

## Abstract

Bone fragility is a common complication in subjects with type 2 diabetes mellitus (T2DM). However, traditional techniques for the evaluation of bone fragility, such as dual-energy X-ray absorptiometry (DXA), do not perform well in this population. Moreover, the Fracture Risk Assessment Tool (FRAX) usually underestimates fracture risk in T2DM. Importantly, novel technologies for the assessment of one microarchitecture in patients with T2DM, such as the trabecular bone score (TBS), high-resolution peripheral quantitative computed tomography (HR-pQCT), and microindentation, are emerging. Furthermore, different serum and urine bone biomarkers may also be useful for the evaluation of bone quality in T2DM. Hence, in this article, we summarize the limitations of conventional tools for the evaluation of bone fragility and review the current evidence on novel approaches for the assessment of quality and bone microstructure alterations in patients with T2DM.

## 1. Introduction

In the last few decades, type 2 diabetes mellitus (T2DM) has dramatically increased in prevalence worldwide, resulting in significant burdens on patients suffering from this condition and healthcare systems [1]. Of note, the rising prevalence of this disease is associated with the development of a wide range of complications, including retinopathy, nephropathy, neuropathy, and cardiovascular disease [1,2]. These complications often affect the quality of life of patients with T2DM, including their physical and psychological functioning [3]. Although some of these comorbidities have a well-known impact on the quality of life [4,5], others have received less attention [6].

Mounting evidence reveals that bone fragility is common in T2DM [7]. Several studies have shown that T2DM constitutes an independent risk factor for osteoporotic fractures, presenting a particularly strong association with hip fractures [8,9,10,11]. Indeed, a number of meta-analyses have confirmed that T2DM is associated with an increased risk of incident hip, vertebral, and non-vertebral fractures [12,13,14]. Since T2DM has a strong relationship with hip fractures that need replacement surgery using total hip arthroplasty, new techniques have been developed in this field [15,16]. Importantly, increases in the incidence of fractures lead to greater costs and healthcare resource utilization in this population [17]. Moreover, fractures are associated with functional impairment and reduction of health-related quality of life [18,19]. Given the important health and socioeconomic impact of skeletal fragility and fractures, individuals with T2DM, especially those with major diabetes-related determinants and other conventional risk factors for osteoporosis, should be assessed for the presence of bone fragility and their fracture risk [20]. However, traditional imaging techniques and fracture risk assessment tools may not be accurate for this purpose in patients with T2DM [21]. 

In this review, we summarize the main limitations of commonly used methods to evaluate bone fragility and estimate fracture risk in patients with T2DM, and we also discuss the potential role of novel strategies in the evaluation of quality and bone microstructure alterations in this population. Although some of these issues have been addressed in previous works [22], the current knowledge on novel techniques and biomarkers for the evaluation of bone fragility in T2DM is still limited. We have updated all the information available on the pathogenic mechanisms that explain bone fragility in patients with T2DM. In addition, we have reviewed the role of new technologies and biomarkers in the assessment of bone fragility in T2DM, considering the main clinical studies currently available.

## 2. Search Strategy and Limitations of the Review

We conducted a comprehensive literature search of articles published in PubMed until March 2022. Peer-reviewed articles related to T2DM and bone fragility published in English were selected, with special attention to clinical studies evaluating bone mineral density (BMD) by dual energy X-ray absorptiometry (DXA) in patients with T2DM, as well as clinical studies assessing bone microstructure through the trabecular bone score (TBS), high-resolution peripheral quantitative computed tomography (HR-pQCT), and microindentation in this population. Finally, we included clinical studies related to the evaluation of novel non-invasive biomarkers of bone quality and fracture risk prediction in T2DM. Original human research articles, including randomized controlled trials, prospective and retrospective observational studies, and cross-sectional studies were considered. The largest studies, as well as the most recent and solid available evidence, were prioritized. Remarkably, a considerable number of the available studies were conducted in postmenopausal women with T2DM; therefore, these results have to be considered cautiously in subjects with T2DM and different characteristics. Moreover, several studies included in this review had a cross-sectional design; thus, further large-scale long-term prospective studies are needed in this field.

## 3. Determinants of Skeletal Fragility and Increased Risk of Fracture in T2DM

Several determinants have been identified in the pathogenesis of bone fragility and increased fracture risk in subjects with T2DM [23] (Figure 1). Notably, a longer duration of T2DM was reported to be an independent risk factor for major osteoporotic fractures in women aged ≥40 and with ≥10 years of diabetes duration [24], and a recent meta-analysis showed a greater increase in the risk of both hip and non-vertebral fractures in subjects with longer diabetes duration [13]. Besides this, poor glycemic control is closely linked to fracture risk, as several large-scale population-based cohort studies have demonstrated [25,26,27]. In this regard, the generation of advanced glycation end-products (AGEs) resulting from chronic exposure to hyperglycemia is one of the key mechanisms in the pathophysiology of bone fragility in T2DM [23]. As such, non-enzymatic glycosylation of collagen leads to the formation of collagen-AGEs, which are involved in the development of impaired bone mineralization and quality through different alterations of the extracellular matrix, a reduction of alkaline phosphatase activity in osteoblasts, and an overactivation of the receptor for AGEs (the latter associated with the release of pro-inflammatory cytokines and reactive oxygen species—ROS—by osteoclasts) [23,28]. On the other hand, it is also postulated that the main event related to bone fragility in T2DM is an overall inhibition of bone cells function and decreased bone turnover [23,29]. This effect may be driven in part by insulin resistance [30].

In addition to chronic hyperglycemia and AGE formation, other mechanisms play a role in bone fragility in T2DM, as previously reviewed [7,23,31]. Among them, a pro-inflammatory state and oxidative stress, along with adipokine dysregulation and marrow adiposity, have a strong influence on bone metabolism [7,31]. Loss of incretin effect has also been implicated in the pathogenesis of skeletal fragility in T2DM [31,32]. Microvascular disease and impaired vascular bone intercommunication determine alterations of bone quality and microarchitecture [7,31]. Ischemic heart disease has also been reported to be associated with an increased risk of vertebral fractures in T2DM [33]. Vitamin D deficiency, commonly found in patients with T2DM, could play a role in both T2DM development and bone fragility [34]. Pathological changes in gut microbiota composition in T2DM may also trigger bone alterations in this population [35].

Further to this, glucose-lowering agents may also be crucial contributors to the reported associations between T2DM and bone fragility [36,37]. The potential benefits of some drugs for bone density and fracture risk (i.e., metformin, glucagon-like peptide 1 receptors agonists and dipeptidyl peptidase-4 inhibitors) [38,39,40] remain to be confirmed in specifically designed studies. Conversely, the long-term use of thiazolidinediones has been independently associated with fracture risk [41], and sodium-glucose cotransporter-2 inhibitors could also have this effect [42,43]. Remarkably, both insulin and sulfonylureas significantly increase fall-related fractures due to episodes of hypoglycemia [44]. In this vein, other prevalent factors in T2DM (i.e., visual impairment, peripheral neuropathy, autonomic dysfunction/postural hypotension, foot ulcers/amputation, and sarcopenia) also lead to an increased risk of fall-related fractures [31,45].

## 4. Bone Density and Fracture Risk Prediction in T2DM

Despite skeletal fragility and fracture risk being greater in subjects with T2DM, this condition is usually associated with normal or even increased BMD measured by DXA [46]. Thus, women with T2DM in the Women’s Health Initiative Observational Study presented higher hip and spine BMD scores compared to those without T2DM [47]. Similarly, in a cross-sectional study including two Swedish cohorts, both men and women exhibited a progressively higher hip BMD according to normal fasting plasma glucose/impaired fasting plasma glucose/T2DM subgroups [48]. In the prospective population-based cohort from the Rotterdam Study, inadequate glycemic control was associated with both higher BMD and increased fracture risk in participants with T2DM [27]. Furthermore, a meta-analysis of 15 observational studies (3473 subjects with T2DM and 19,139 healthy controls) showed that participants with T2DM had significantly higher BMD at the femoral neck, hip, and spine [49]. 

It is noteworthy that these results contrast with those reported by studies assessing BMD in type 1 diabetes mellitus (T1DM), in which BMD is generally low [50]. Although the mechanisms involved in the association between T2DM and normal/high BMD are not fully understood, some data suggest that these findings might be related to chronic hyperinsulinemia and insulin resistance [51], as well as the effect of some adipokines, such as leptin, on bone metabolism [52]. Excess weight/obesity, which are often encountered in patients with T2DM, could also play a role in increased BMD, although some studies have reported that this relationship remains after adjusting for the body mass index (BMI) [49]. Since T2DM is associated with increased fracture risk, regardless of whether there is a normal/high BMD, a fact known as “the diabetic paradox of bone fragility” [53], the diagnosis of osteoporosis based on BMD measured by DXA, should be cautiously considered [21].

On the other hand, the Fracture Risk Assessment Tool (FRAX), which is widely used to estimate 10-year absolute fracture risk, has been demonstrated to underestimate the risk for both hip and major osteoporotic fractures in patients with T2DM [54]. These results are influenced, in part, by the higher BMD observed in patients with T2DM [49]. Indeed, contrary to T1DM, T2DM is not included in the FRAX tool as a secondary cause of osteoporosis [55]. In this regard, some authors have proposed a correction factor with the use of glycated hemoglobin in order to improve the predictive ability of this algorithm for fracture risk [56]. Recently, adjustment of FRAX for T2DM has been suggested in order to create a useful alternative [57,58], although further research is warranted to confirm these results. Alternatively, certain methods (i.e., inputting rheumatoid arthritis, adjusting FRAX by TBS, reducing the femoral T-score by 0.5, and increasing the age by 10 years) have been proposed to improve the performance of FRAX in T2DM, although no single method appears to be optimal in all settings [59]. In light of the above, new approaches to the evaluation of bone fragility in patients with T2DM are needed.

## 5. Bone Microstructure in T2DM

As previously discussed, patients with T2DM have normal or elevated BMD; however, bone microarchitecture alterations may be present in this group, resulting in an increased fracture risk [60]. In this context, the trabecular bone score, high-resolution peripheral quantitative computed tomography, and microindentation are useful techniques for the evaluation of the bone microstructure in T2DM.

### 5.1. Trabecular Bone Score

The TBS is a non-invasive, indirect index of trabecular microarchitecture [61]. It is derived from experimental variograms of the projected two-dimensional lumbar spine DXA image and can assess pixel gray-level variations of this area, which translate into a bone microstructure-related score [61]. Accordingly, a high TBS is related to numerous, well-connected and less sparse trabeculae (i.e., normal bone microarchitecture), whereas a low TBS indicates a reduced number of trabeculae and less connectivity, as well as trabecular separation (i.e., altered bone microarchitecture) [61], as shown in Figure 2. In this regard, the proposed TBS cut-off values are as follows: TBS > 1.31 (normal microarchitecture), TBS between 1.23 and 1.31 (partially degraded microarchitecture), and TBS < 1.23 (degraded microarchitecture) [62].

TBS has been demonstrated to be an independent predictor for osteoporotic fractures [62,63,64]. In addition to this, TBS can detect differences between DXA images with similar BMDs [61] and helps to improve the performance of BMD assessed by DXA in the prediction of osteoporotic fractures [65,66]. Indeed, TBS has been incorporated into the FRAX algorithm (FRAX adjusted for TBS), although the clinical impact of this adjustment is yet to be properly evaluated [62].

In patients with T2DM, TBS has been reported to be significantly decreased compared to subjects without diabetes, which suggests that this index could be a useful tool for the diagnosis of bone fragility in this population [67]. TBS may be decreased even in prediabetes, indicating that the degradation of bone microarchitecture may occur in early stages of the disease [68]. Interestingly, in a recent cross-sectional study including 137 patients with T2DM aged 49–85 and 300 healthy controls, the presence of T2DM was associated with significantly lower TBS values despite higher lumbar spine BMD; adiposity (estimated by the relative fat mass) and insulin resistance could play a role in these results [69]. Accordingly, visceral fat reduction may increase TBS values [70]. Furthermore, higher glycated hemoglobin levels and a longer disease duration in patients with T2DM are related to lower TBS values, although the interference of abdominal soft tissue thickness should be considered when interpreting these findings [68,71,72,73]. Moreover, diabetic microvascular disease may be linked to lower TBS [74]. 

Notably, several studies have shown that TBS can predict incident/prevalent osteoporotic fractures independent of BMD [75,76,77,78] (Table 1). In a retrospective cohort study from the Manitoba Bone Density Program (29,407 women ≥ 50 years, 2356 with diagnosed T2DM), lumbar spine TBS was a BMD-independent predictor of major osteoporotic fractures in both participants with and without T2DM [75]. In a study including 206 postmenopausal women with/without T2DM, TBS values ≤1.130 presented an adequate diagnostic accuracy for vertebral fractures in the former [76], whereas, in a cross-sectional study conducted on 548 patients with T2DM, TBS correlated with prevalent vertebral fractures [77]. Finally, in a study including 285 postmenopausal women with T2DM, TBS had the strongest association with vertebral fractures [78]. Considering all these findings together, TBS may constitute a useful approach for the diagnosis of bone fragility and the evaluation of fracture risk in T2DM, although further prospective studies are needed to corroborate these data.

### 5.2. High-Resolution, Peripheral, Quantitative Computed Tomography

HR-pQCT is a non-invasive three-dimensional imaging modality that permits the assessment of bone microarchitecture, including the measurement of volumetric cortical and trabecular bone mineral density (vBMD), cortical thickness/porosity, bone strength, and other parameters in the appendicular skeleton (i.e., distal radius and tibia) [79]. In recent years, HR-pQCT has emerged as a promising technique that could become widely used for the diagnosis of osteoporosis and for clinical fragility fracture prediction [80,81]. 

In a pilot study conducted on 19 postmenopausal women with T2DM matched to 19 controls, Burghardt et al. showed for the first time that T2DM may be associated with bone microarchitecture alterations, as assessed by HR-pQCT [82]. It was observed that, although participants with T2DM had higher trabecular vBMD and trabecular thickness, they also presented higher cortical porosity and impaired bone strength, measured by micro-finite element analysis [82]. Similarly, Patsh et al. reported increased cortical porosity at the ultradistal and distal radio and tibia in 80 postmenopausal women with T2DM [83], while Yu and colleagues also found defects in cortical bone microarchitecture (i.e., higher cortical porosity and lower cortical vBMD) in African American women with T2DM compared to healthy controls [84]. Data from the Framingham Study (a total of 1069 subjects underwent HR-pQCT, 129 subjects with T2DM) showed that patients with T2DM had lower vBMD and higher cortical porosity compared to controls [85]. Interestingly, in a prospective exploratory study that involved postmenopausal women with T2DM with/without a history of fragility fractures and controls, patients with T2DM and a history of fractures exhibited the highest cortical porosity [86]. Cortical porosity increased over time similarly in the three groups, although patients with T2DM and a history of fractures presented the greatest decreases in bone strength indices in the follow-up period, a fact that suggests that cortical porosity may develop early, followed by small increases in this parameter along with significant material strength impairment [86]. Of note, cortical bone deficits assessed by HR-pQCT in T2DM may be driven by the presence of microvascular disease and/or poor metabolic control [87,88].

Conversely, other studies did not find significant differences in bone microarchitecture determined by HR-pQCT between subjects with and without T2DM [89]. Intriguingly, in a population-based sample of women aged 75–80 (99 women with T2DM and 954 controls), T2DM was associated with better bone microarchitecture (including higher trabecular and cortical vBMD in several regions and lower cortical porosity) [90]. In this context, large-scale clinical studies on the topic are required to evaluate the role of HR-pQCT in the diagnosis of bone fragility in T2DM. Moreover, the impacts of cortical porosity and other parameters, as estimated by HR-pQCT, on the prediction of fractures in T2DM are yet to be elucidated.

### 5.3. Microindentation

Microindentation is an invasive technique that enables percutaneous evaluation of the resistance of bone to indentation in vivo [91]. By indenting a probe tip through the skin covering the tibia and measuring the depth that it penetrates the bone after the generation of an impact force, impact microindentation measurement directly assesses the mechanical characteristics of cortical bone, which are estimated by the bone material strength index (BMSi) [92]. This technique may be particularly useful in populations presenting discrepancies between BMD and increased fracture risk, such as those with T2DM [93]. Accordingly, some studies have reported decreased BMSi in postmenopausal women with T2DM [89,90,94]. Moreover, altered matrix bone properties evaluated by microindentation were confirmed in this population, even though BMD assessed by DXA and/or bone microarchitecture assessed by HR-pQCT showed no differences between subjects with T2DM and healthy controls [89,90]. Remarkably, in a cross-sectional study including 340 men aged 33–96, participants with T2DM exhibited lower mean BMSi compared to subjects with normoglycemia/impaired fasting glucose [95]. However, it should be noted that further work is needed with regard to this technique for the assessment of bone fragility in patients with T2DM.

## 6. Bone Quality in T2DM: The Role of Biomarkers of Bone Fragility

In addition to bone mineralization and microarchitecture, skeletal material properties are also influenced by bone turnover and the quality of collagen, which may be affected by the accumulation of AGEs, leading to the alteration of collagen crosslinks and function as discussed in previous sections [23]. In this regard, it has been stated that bone turnover is decreased in T2DM, which results in reduced serum levels of bone remodeling markers [23,96,97,98]. However, it remains unknown whether these biochemical markers may be helpful for the diagnosis of bone fragility or the prediction of fracture risk in patients with T2DM. On the one hand, decreased circulating levels of parathyroid hormone (PTH) along with osteocalcin were shown to be associated with a higher risk of vertebral fracture in postmenopausal women with T2DM [99]. On the contrary, in a recent study, Napoli et al. showed that serum bone turnover markers (terminal telopeptide of type 1 collagen-CTX, osteocalcin, and procollagen type 1 N-terminal propeptide-P1NP) were not able to predict fracture risk in T2DM [100].

On the other hand, AGES related to collagen, such as pentosidine and N-carboxymethyl lysine (CML), are increased in bone biopsy specimens from subjects with T2DM [60,101,102]. Therefore, circulating/urinary levels of these AGEs may become attractive surrogate markers of bone quality in subjects with T2DM. Besides this, other novel biomarkers could play a role in the evaluation of bone fragility in T2DM.

### 6.1. Pentosidine

Pentosidine is a well-characterized AGE derived from the non-enzymatic reaction of pentoses with lysine and arginine residues [103]. Pentosidine levels are increased in T2DM [104]; moreover, circulating levels of pentosidine appear to be higher in patients with T2DM and poor metabolic control, and they are also related to T2DM-associated cardiovascular disease and microvascular complications [104,105,106].

Higher concentrations of pentosidine can also be found in the cancellous bone of patients with T2DM, and this accumulation may be associated with bone fragility via reduced post-yield strain and toughness due to alterations of the bone matrix [60,107,108]. These disturbances may be related to a decreased bone turnover induced by this AGE [109]. Of note, serum/urinary levels of pentosidine may also be applicable markers of bone fragility in T2DM. Thus, serum levels of pentosidine have been reported to be linked to the presence of vertebral fractures in postmenopausal women with T2DM, who presented similar BMD values/bone turnover markers to controls [110]. Furthermore, in a cross-sectional study, urine pentosidine levels were higher in patients with T2DM and vertebral fractures, and were negatively correlated with TBS [111]. In an observational cohort study (501 participants with T2DM and 427 without T2DM), Schwartz et al. showed that urine pentosidine was able to predict incident clinical fractures only in adults with T2DM, while prevalent vertebral fractures were also associated with urine pentosidine in this population [112].

### 6.2. N-carboxymethyl Lysine

The AGE N-carboxymethyl lysine (CML) may also play an important role in bone fragility in patients with T2DM [102]. In this regard, CML content in human cortical bone has been reported to be higher in subjects with T2DM, which may affect collagen properties [102]. In a large cohort from the Cardiovascular Health Study (3373 participants), serum levels of CML were associated with increased risk of incident hip fracture, independent of the BMD, with no differences in the hazard ratio between participants with and without T2DM [113]. Recently, in a cohort study including 712 participants with T2DM and 2332 subjects without, Dhaliwal et al. showed that circulating levels of CML were higher in patients with T2DM, and higher levels of this AGE were related to an increased risk of incident clinical fractures in this group, independent of the BMD [114]. Indeed, in this study, no relationship was found between hip BMD and CML, which reinforces the notion that bone quality is a major determinant of the pathophysiology of increased fracture risk in T2DM [114].

### 6.3. Sclerostin

Sclerostin is an inhibitor of the pro-osteogenic Wnt signaling pathway, which results in decreased bone turnover [23,115]. Hence, some studies have found that higher levels of this protein could be associated with a higher risk of osteoporotic fractures [116,117].

Increased circulating levels of sclerostin have been observed in patients with T2DM and may be involved in low bone turnover and a greater risk of fracture found in this population [118]. Thus, higher serum levels of sclerostin have been reported in postmenopausal women with T2DM and fragility fractures, compared to those without fragility fractures [119,120]. In addition to this, in a cross-sectional study including postmenopausal women and men aged >50 years with T2DM, elevated sclerostin levels correlated with the presence of vertebral fractures [121].

### 6.4. MicroRNAs

MicroRNAs (miRNAs) are epigenetic regulators of different cellular processes, including bone development, homeostasis, and healing [122]. Although evidence regarding the role of these elements in bone fragility in T2DM is still limited, some studies have shed light on their potential utility [123,124,125]. In a study conducted on 168 postmenopausal women with T2DM, three different miRNAs, including senescent miR-31-5p, were significantly associated with incident fragility fractures [123]. In previous analyses, Heilmeier et al. also reported that individual miRNAs or miRNA combinations were able to discriminate the fracture status in postmenopausal women with T2DM [124]. Chen et al. also described several miRNAs with potential implications for fracture prediction in postmenopausal women with T2DM [125].

### 6.5. Other Biomarkers

Aside from in the serum and urine, AGE deposition can be measured in other tissues, such as the skin. Therefore, skin autofluorescence (SAF), which is based on the non-invasive measurement of AGE accumulation in the human skin, has emerged as a promising technique [126]. However, little evidence is available concerning bone fragility/fracture risk estimation through this tool. In two cross-sectional studies, SAF was inversely correlated with BMSi in patients with T2DM [94,127]. Interestingly, SAF was associated with prevalent vertebral and major osteoporotic fractures in participants from the Rotterdam Study [128]. However, these data must be assessed specifically in individuals with T2DM.

In another area, the fingernail quality may serve as a non-invasive marker of the bone quality in T2DM [129,130]. Nevertheless, further investigation is needed.

## 7. Conclusions

Since traditional methods for the evaluation of BMD and fracture risk in individuals with T2DM can lead to significant errors, additional techniques are needed. TBS may be considered as a useful non-invasive index of bone microarchitecture, which is often altered in patients with T2DM. Since TBS is derived from DXA images, it may represent an applicable tool for the diagnosis of bone fragility in T2DM. In addition, it could facilitate follow-up and the evaluation of response to treatment in these patients, and may help to unravel the role of certain glucose-lowering agents in bone fragility. HR-pQCT also permits the evaluation of bone microstructure; however, this technique involves significant costs and exposure to radiation, which should be considered. Future opportunities in this area include the evaluation of bone microstructure by DXA-3D, which has shown remarkable results in several conditions other than T2DM and may provide accurate estimations of bone structure and strength, thus offering additional information with regard to fracture risk. Despite the fact that microindentation is a promising method for the evaluation of bone matrix properties, it requires an invasive procedure, which may limit its application in clinical practice. On the other hand, some biochemical markers may represent interesting non-invasive alternatives for the evaluation of skeletal fragility/fracture risk prediction in patients with T2DM, although it is noteworthy that the current evidence regarding some of these alternatives is still limited; therefore, further research (e.g., validation studies) is needed before these biomarkers may be included in routine practice. Further large-scale, long-term prospective studies are needed in the evaluation of quality and bone microstructure alterations in patients with T2DM.

## Figures and Tables

**Figure 1 jcm-11-02206-f001:**
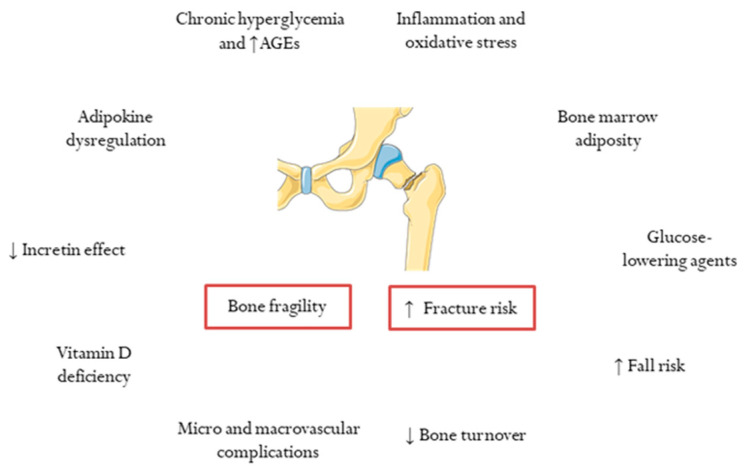
Determinants of bone fragility and increased fracture risk in type 2 diabetes. AGEs, advanced glycation end products.

**Figure 2 jcm-11-02206-f002:**
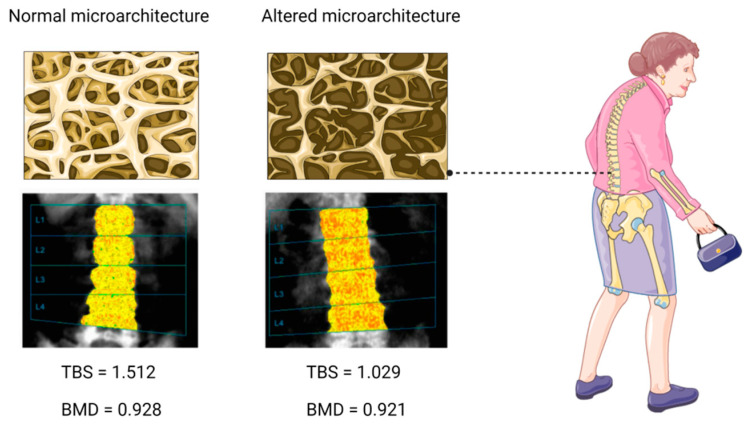
Trabecular bone score (TBS) as a useful technology for the assessment of the trabecular microarchitecture. TBS > 1.31 (**left**) denotes a normal microarchitecture, whereas TBS < 1.23 (**right**) indicates an altered microarchitecture. TBS can detect differences between similar values of lumbar spine bone mineral density (BMD) estimated by dual-energy X-ray absorptiometry (DXA) (g/cm^2^).

**Table 1 jcm-11-02206-t001:** Clinical studies showing an independent association between the trabecular bone score (TBS) and osteoporotic fractures in patients with type 2 diabetes mellitus.

Study	Design	Study Population	Results
Leslie et al., 2013 [75]	Retrospective cohort (mean follow-up 4.7 years)	29,407 women ≥ 50 years (2356 with diagnosed T2DM)	TBS predicted major osteoporotic fractures (hip, spine, forearm and humerus) in T2DM (HR 1.27, CI 1.10–1.46)
Zhukouskaya et al., 2015 [76]	Cross-sectional	99 postmenopausal women with T2DM/107 healthy controls	TBS was associated with VF (AUC 0.69, cut-off value 1.130 in ROC curve analysis)
Yamamoto et al., 2019 [77]	Cross-sectional	584 patients with T2DM (257 postmenopausal women and 291 men > 50 years)	TBS correlated with prevalent VF in multivariate logistic regression analysis
Lin et al., 2019 [78]	Cross-sectional	285 postmenopausal women with T2DM	TBS had the strongest association with VF (AUC 0.775)

T2DM, type 2 diabetes mellitus; TBS, trabecular bone score; VF, vertebral fractures; HR, hazard ratio; CI, confidence interval; ROC, receiver operating characteristic; AUC, area under the curve.

## Data Availability

Not applicable.

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
