# Peer review of "Evaluation of Quality and Bone Microstructure Alterations in Patients with Type 2 Diabetes: A Narrative Review"

_jcm, 2022, doi:10.3390/jcm11082206_

Round 1
Reviewer 1 Report
The article entitled “Evaluation of quality and bone microstructure alterations in patients with type 2 diabetes” aimed to analyze the limitations of conventional tools for the evaluation of bone fragility and review the current evidence on novel approaches for the assessment of quality and bone microstructure alterations in patients with T2DM. The paper is in line with journal’s aim, moreover, Authors have well revised several issues; however, I ask authors to add some key concepts.
- In the title of the article, authors should specify the type of study (Is it a narrative review?)
- The abstract section is confusing and too long, please rewrite the entire paragraph
- The introduction section is too short, the author must add and discuss previous findings regarding DM and its most frequent complications affect the life quality of sufferers (please see and discuss doi: 10.3390 / jcm9072196)
- It would be necessary to introduce a paragraph concerning the research methodology, at least it would be useful to know the criteria for inclusion and exclusion of the articles selected for the overview
- The limits of the study should be included in the paper
- Conclusions cannot be reduced to a sentence: you must improve them highlighting the limits and the future insights pointed out from this article.
According to this Reviewer’s consideration, novelty and quality of the paper, publication of the present manuscript is recommended after minor revision.
Author Response
Dear Editor and Reviewers,
We would like to thank you for your constructive comments and recommendations, which have undoubtedly helped us to improve our manuscript. We have taken these suggestions into consideration and have revised the paper accordingly. We have made all possible efforts to respond to each of the reviewers’ comments and have edited the manuscript to address the reviewers’ suggestions fully. Moreover, we have edited the English using MDPI editing service.
We have provided the replies to the comments in the following section and have highlighted changes in the manuscript in a red font.
We hope that the revised version of our manuscript may now be found acceptable for publication in Journal of Clinical Medicine. Nevertheless, we are willing to revise it further according to any other suggestions or concerns raised by the Editor or the Reviewers.
Yours faithfully,
The authors
Comments and Suggestions for Authors
Reviewer 1
Comment 1: The article entitled “Evaluation of quality and bone microstructure alterations in patients with type 2 diabetes” aimed to analyse the limitations of conventional tools for the evaluation of bone fragility and review the current evidence on novel approaches for the assessment of quality and bone microstructure alterations in patients with T2DM. The paper is in line with journal’s aim, moreover, Authors have well revised several issues; however, I ask authors to add some key concepts.
Response 1: First of all, we thank the reviewer’s comment and his/her constructive review of our manuscript. We appreciate that the reviewer thinks that the paper is in line with journal’s aim and that the authors have well revised several issues.
Comment 2: In the title of the article, authors should specify the type of study (Is it a narrative review?).
Response 2: We thank the reviewer for his/her observation. Following the reviewer’s recommendation, we have specified the type of the study in the title of the article as follows: “Evaluation of Quality and Bone Microstructure Alterations in Patients with Type 2 Diabetes: A Narrative Review”
Comment 3: The abstract section is confusing and too long, please rewrite the entire paragraph
Response 3: Thank you for your helpful recommendation. As suggested, we have modified this section in order to improve readability as follows:
“Bone fragility is a common complication in subjects with type 2 diabetes mellitus (T2DM). However, traditional techniques for the evaluation of bone fragility, such as dual-energy X-ray absorptiometry (DXA), do not perform well in this population. Also, the Fracture Risk Assessment Tool (FRAX) usually underestimates the fracture risk in T2DM. Importantly, novel technologies for the assessment of the bone microarchitecture in patients with T2DM, such as the trabecular bone score (TBS), high-resolution peripheral quantitative computed tomography (HR-pQCT) and microindentation, are emerging. Furthermore, different serum and urine bone biomarkers may also be useful for the evaluation of the bone quality in T2DM. Hence, in this article, we summarize the limitations of conventional tools for the evaluation of bone fragility and review the current evidence on novel approaches for the assessment of the quality and bone microstructure alterations in patients with T2DM.”
Comment 4: The introduction section is too short, the author must add and discuss previous findings regarding DM and its most frequent complications affect the life quality of sufferers (please see and discuss doi: 10.3390 / jcm9072196)
Response 4: Thanks for your interesting comment. As recommended, we have included additional information in this section in order to discuss previous findings regarding DM and how its most frequent complications affect the quality of life, including the aforementioned citation:
“…These complications often affect the quality of life of patients with T2DM, including their physical and psychological functioning [3]. Although some of these comorbidities have a well-known impact on the quality of life [4,5], others have received less attention [6].…”
“…Besides, fractures are associated with functional impairment and reduction of health-related quality of life [18,19]...”
Comment 5: It would be necessary to introduce a paragraph concerning the research methodology, at least it would be useful to know the criteria for inclusion and exclusion of the articles selected for the overview
Response 5: Thanks for this remark. As indicated, we have included a paragraph concerning the research methodology. Please, take into account that a new section has been included in this regard (“2. Search Strategy and Limitations of the Review”) including the following information:
“We conducted a comprehensive literature search of articles published in Pubmed until March 2022. Peer-reviewed articles related to T2DM and bone fragility published in English were selected, with special attention to clinical studies evaluating bone mineral density (BMD) by dual energy X-ray absorptiometry (DXA) in patients with T2DM, and clinical studies assessing bone microstructure through trabecular bone score (TBS), high-resolution peripheral quantitative computed tomography (HR-pQCT) and microindentation in this population. Finally, we included clinical studies related to the evaluation of novel non-invasive biomarkers of bone quality and fracture risk prediction in T2DM. Original human research articles, including randomized controlled trials, prospective and retrospective observational studies and cross-sectional studies were considered. The largest studies, as well as the most recent and solid available evidence were prioritized. Remarkably, a considerable number of the available studies were conducted in postmenopausal women with T2DM; therefore, these results have to be considered cautiously in subjects with T2DM and different characteristics. Also, several studies included in this review had a cross-sectional design; thus, further large-scale long-term prospective studies are needed in this field.”
Comment 6: The limits of the study should be included in the paper.
Response 6: Thanks you for this comment. Following the reviewer’s recommendation, we have remarked the limits of the study in the revised version of the manuscript. Please, take into account that all changes can be found along the manuscript as follows:
Search Strategy and Limitations of the Review Section: “…Remarkably, a considerable number of the available studies were conducted in postmenopausal women with T2DM; therefore, these results have to be considered cautiously in subjects with T2DM and different characteristics. Also, several studies included in this review had a cross-sectional design; thus, further large-scale long-term prospective studies are needed in this field.…”
Conclusion Section: “…although it is noteworthy that the current evidence regarding some of these alternatives is still limited; therefore, further research (e.g., validation studies) is needed before these biomarkers may be included in routine practice. Further large-scale, long-term prospective studies are needed in the evaluation of the quality and bone microstructure alterations in patients with T2DM.”
Comment 7: Conclusions cannot be reduced to a sentence: you must improve them highlighting the limits and the future insights pointed out from this article.
Response 7: Thanks for this remark. As indicated, we have revised and edited the Conclusion Section in order to highlight the limits and future insights pointed out from this article.
Comment 8: According to this Reviewer’s consideration, novelty and quality of the paper, publication of the present manuscript is recommended after minor revision.
Response 8: We would thank the reviewer for his kind evaluation of our manuscript, considering the novelty and quality of this paper and recommending its publication after minor revision. Thank you for your feedback.
Reviewer 2 Report
Dear Martínez-Montoro et al.,
The manuscript “Evaluation of quality and bone microstructure alterations in patients with type 2 diabetes” (jcm-1643147) by Martínez-Montoro et al. summarize the limitations of conventional tools for the evaluation of bone fragility and review the current evidence on novel approaches for the assessment of quality and bone microstructure alterations in patients with T2DM.. The topic is interesting, but I think this article should reconsider after proper changes in major revision for publication in Journal of Clinical Medicine. Some of my specific comments are:
- To improve the quality of English used in this manuscript and make sure English language, grammar, punctuation, spelling, and overall style are correct, further proofreading is needed. As an alternative, the authors can use the MDPI English proofreading service for this issue. For example, the overused “that” as conjunction was used 36 times in the present manuscript, which makes the manuscript monotonous.
- Please make sure the authors have used the Journal of Clinical Medicine, MDPI format correctly. The authors can download published manuscripts by Journal of Clinical Medicine, MDPI, and compare them with the present author's manuscript to ensure typesetting is appropriate. Some errors are:
- The uppercase and lowercase use in the title (line 2-3) and all of subsection of the present manuscript in incorrect
- Information about Author Contributions that should be put after conclusion section and Funding (around line 352-353) information is missing.
- And others
- In the keywords section (line 28-29), the authors are recommended only to include a maximum of five keywords, where in the current form it has six keywords.
- Describe the novelty of the article made by the author? From the results of my evaluation, it seems that many similar published works adequately explain what you have raised in the current review manuscript related to bone microstructure on the patients with type 2 diabetes as the best reviewer knowledge in this research area, for example:
- Type 2 Diabetes Mellitus Increases the Risk to Hip Fracture in Postmenopausal Osteoporosis by Deteriorating the Trabecular Bone Microarchitecture and Bone Mass. 2019, 3876957. https://doi.org/10.1155/2019/3876957
- Regulation of DMT1 on Bone Microstructure in Type 2 Diabetes. International Journal of Medical Sciences. 2015, 12(5), 441-449. https://doi.org/10.7150/ijms.11986
- Obesity, Type 2 Diabetes and Bone in Adults. Calcified Tissue International. 2017, 100. 528-538. https://doi.org/10.1007/s00223-016-0229-0
- And others
If there is something others really new in this manuscript, please highlight it more clearly in the introduction section (line 31-51).
- State of the art and significance of the present review is not clearly present, the authors should highlight it more advanced in the introduction section (line 31-51). It will explain why this review article needs to exist.
- Since diabetes has a strong relation to hip fractures that need replacement surgery using total hip arthroplasty. I would encourage and advise the authors to adopt some of the additional references related to total hip arthroplasty published by MDPI in the introduction section (line 31-51) as follow:
- Tresca Stress Simulation of Metal-on-Metal Total Hip Arthroplasty during Normal Walking Activity. Materials (Basel). 2021, 14, 7554. https://doi.org/10.3390/ma14247554
- The Effect of Bottom Profile Dimples on the Femoral Head on Wear in Metal-on-Metal Total Hip Arthroplasty. J. Funct. Biomater. 2021, 12, 38. https://doi.org/10.3390/jfb12020038
- To make the reader easier to understand and more interested, the authors are recommended and advised to give an additional figure(s) as illustrative, where in the present manuscript only one figure is available (Figure 1, line 149-153).
- The conclusion (line 332-352) of the present manuscript is too long and not solid. Further elaboration is needed and makes it more concise.
- Further research needs to be explained in the conclusion section (line 332-352).
- Overall, the substance presented in the current review article is still too little, the author needs to enhance the discussion in-depth, so it can improve the quality of the existing review articles and giving more serious scientific contributions.
I am pleased to have been able to review the author's present manuscript. Hopefully, the author can revise the current manuscript as well as possible so that it becomes even better. Good luck for the author's work and effort.
Best regards,
The Reviewer
Author Response
Dear Editor and Reviewers,
We would like to thank you for your constructive comments and recommendations, which have undoubtedly helped us to improve our manuscript. We have taken these suggestions into consideration and have revised the paper accordingly. We have made all possible efforts to respond to each of the reviewers’ comments and have edited the manuscript to address the reviewers’ suggestions fully. Moreover, we have edited the English using MDPI editing service.
We have provided the replies to the comments in the following section and have highlighted changes in the manuscript in a red font.
We hope that the revised version of our manuscript may now be found acceptable for publication in Journal of Clinical Medicine. Nevertheless, we are willing to revise it further according to any other suggestions or concerns raised by the Editor or the Reviewers.
Yours faithfully,
The authors.
Comments and Suggestions for Authors
Reviewer 2
Comment 1: The manuscript “Evaluation of quality and bone microstructure alterations in patients with type 2 diabetes” (jcm-1643147) by Martínez-Montoro et al. summarize the limitations of conventional tools for the evaluation of bone fragility and review the current evidence on novel approaches for the assessment of quality and bone microstructure alterations in patients with T2DM. The topic is interesting, but I think this article should reconsider after proper changes in major revision for publication in Journal of Clinical Medicine. Some of my specific comments are.
Response 1: First of all, we thank the reviewer’s comment and his/her constructive review of our manuscript, also considering that the topic of this review is interesting.
Comment 2: To improve the quality of English used in this manuscript and make sure English language, grammar, punctuation, spelling, and overall style are correct, further proofreading is needed. As an alternative, the authors can use the MDPI English proofreading service for this issue. For example, the overused “that” as conjunction was used 36 times in the present manuscript, which makes the manuscript monotonous.
Response 2: We appreciate your suggestion. According to the reviewer´s recommendation, we have used the MDPI English editing service to improve the quality of English in the revised version of the manuscript.
Comment 3: Please make sure the authors have used the Journal of Clinical Medicine, MDPI format correctly. The authors can download published manuscripts by Journal of Clinical Medicine, MDPI, and compare them with the present author's manuscript to ensure typesetting is appropriate. Some errors are: The uppercase and lowercase use in the title (line 2-3) and all of subsection of the present manuscript in incorrect. Information about Author Contributions that should be put after conclusion section and Funding (around line 352-353) information is missing. In the keywords section (line 28-29), the authors are recommended only to include a maximum of five keywords, where in the current form it has six keywords.
Response 3: Thank you for your helpful comments and suggestions. Following the reviewer´s advice, we have modified the manuscript in order to address Journal of Clinical Medicine- MDPI format correctly.
Comment 4: Describe the novelty of the article made by the author? From the results of my evaluation, it seems that many similar published works adequately explain what you have raised in the current review manuscript related to bone microstructure on the patients with type 2 diabetes as the best reviewer knowledge in this research area, for example: Type 2 Diabetes Mellitus Increases the Risk to Hip Fracture in Postmenopausal Osteoporosis by Deteriorating the Trabecular Bone Microarchitecture and Bone Mass. 2019, 3876957. https://doi.org/10.1155/2019/3876957. Regulation of DMT1 on Bone Microstructure in Type 2 Diabetes. International Journal of Medical Sciences. 2015, 12(5), 441-449. https://doi.org/10.7150/ijms.11986. Obesity, Type 2 Diabetes and Bone in Adults. Calcified Tissue International. 2017, 100. 528-538. https://doi.org/10.1007/s00223-016-0229-0. And other. If there is something others really new in this manuscript, please highlight it more clearly in the introduction section (line 31-51).
Response 4: We thank the reviewer’s comment. As indicated by the reviewer, we have addressed this point in the Introduction Section of the revised manuscript as follows:
“Although some of these issues have been addressed in previous works [22], the current knowledge on novel techniques and biomarkers for the evaluation of bone fragility in T2DM is still limited. We have updated all the information available on the pathogenic mechanisms that explain bone fragility in patients with T2DM. In addition, we have re-viewed the role of new technologies and biomarkers in the assessment of bone fragility in T2DM, considering the main clinical studies currently available.”
Please, take into account that the main novelties related to new technologies and novel biomarkers in the assessment of bone fragility in T2DM (e.g., recently postulated biomarkers, such as N-carboxymethyllysine or microRNAs; physiopathological mechanisms involved in bone alterations assessed by these techniques, as well as the role of chronic complications associated with T2DM in these results; clinical studies comparing the performance between these novel diagnosis tools in patients with T2DM) can be found along the manuscript and are marked with red text. Below, some examples of the discussed novel findings and references included in the text are indicated:
Hayón-Ponce M et al. DOI:10.1016/j.diabet.2021.101276; Dhaliwal R et al. DOI:10.1002/jbmr.4466; Sihota P et al . DOI:10.1371/journal.pone.0257955; Waqas K et al. DOI:10.1002/jbmr.4096; Heilmeier U et al. DOI:10.1016/j.bone.2021.116308
Comment 5: State of the art and significance of the present review is not clearly present, the authors should highlight it more advanced in the introduction section (line 31-51). It will explain why this review article needs to exist.
Response 5: Thanks for this remark. As indicated above, we have included the significance and novelty of the present review in the Introduction Section. Also, we have highlighted state of the art regarding this topic as follows:
“…Although some of these issues have been addressed in previous works [22], the current knowledge on novel techniques and biomarkers for the evaluation of bone fragility in T2DM is still limited. We have updated all the information available on the pathogenic mechanisms that explain bone fragility in patients with T2DM. In addition, we have re-viewed the role of new technologies and biomarkers in the assessment of bone fragility in T2DM, considering the main clinical studies currently available.…”
Comment 6: Since diabetes has a strong relation to hip fractures that need replacement surgery using total hip arthroplasty. I would encourage and advise the authors to adopt some of the additional references related to total hip arthroplasty published by MDPI in the introduction section (line 31-51) as follow: Tresca Stress Simulation of Metal-on-Metal Total Hip Arthroplasty during Normal Walking Activity. Materials (Basel). 2021, 14, 7554. https://doi.org/10.3390/ma14247554. The Effect of Bottom Profile Dimples on the Femoral Head on Wear in Metal-on-Metal Total Hip Arthroplasty. J. Funct. Biomater. 2021, 12, 38. https://doi.org/10.3390/jfb12020038.
Response 6: We are in agreement with your interesting suggestion. As recommended, we have included and discussed these references in the introduction section as follows:
“…Since T2DM has a strong relationship with hip fractures that need replacement surgery using total hip arthroplasty, new techniques have been developed in this field [15,16]. Importantly, increases in the incidence of fractures lead to greater costs and healthcare resource utilization in this population [17]. Besides, fractures are associated with functional impairment and reduction of health-related quality of life [18,19]…”
Comment 7: To make the reader easier to understand and more interested, the authors are recommended and advised to give an additional figure(s) as illustrative, where in the present manuscript only one figure is available (Figure 1, line 149-153).
Response 7: Thank you for your practical advice. As suggested, we have included a figure summarizing the main determinants of bone fragility and increased fracture risk in type 2 diabetes (Figure 1).
Comment 8: The conclusion (line 332-352) of the present manuscript is too long and not solid. Further elaboration is needed and makes it more concise.
Response 8: Thanks for this remark. As recommended, we have modified this section in the revised version of the manuscript in order to address the reviewer’s suggestion.
Comment 9: Further research needs to be explained in the conclusion section (line 332-352).
Response 9: We thank the reviewer for his/her interesting suggestion. As recommended, we have discussed this point in the revision version of the manuscript. Some examples of new additions to the manuscript are as follows:
“…although it is noteworthy that the current evidence regarding some of these alternatives is still limited; therefore, further research (e.g., validation studies) is needed before these biomarkers may be included in routine practice…”
“…Since the TBS is derived from DXA images it may represent an applicable tool for the diagnosis of bone fragility in T2DM. In addition, it could facilitate follow-up and response to treatment in these patients and may help to unravel the role of certain glucose-lowering agents…”
Comment 10: Overall, the substance presented in the current review article is still too little, the author needs to enhance the discussion in-depth, so it can improve the quality of the existing review articles and giving more serious scientific contributions.
Response 10: Thanks you for your recommendation. As indicated, we have extensively modified the manuscript in order to enhance the main limitations regarding this topic (including a new section “2. Search Strategy and Limitations of the Review”), as well as the current knowledge and future perspectives in this field.
Finally, we would thank the reviewer for his/her constructive evaluation of our manuscript and valuable comments. We expect that the revised version of our manuscript may now be found acceptable for publication. Thank you for your feedback.
Round 2
Reviewer 2 Report
Dear Martínez-Montoro et al.,
After carefully reading the author's revised manuscript entitled "Evaluation of quality and bone microstructure alterations in patients with type 2 diabetes" (jcm-1643147) by Martínez-Montoro et al., The authors have been made significant improvements in the revised manuscript. Also, all of the issues in my review report have been addressed precisely.
With my pleasure, I recommend the manuscript should be accepted for publication on Journal of Clinical Medicine
Best regards,
The Reviewer